# Detection of Highway Pavement Damage Based on a CNN Using Grayscale and HOG Features

**DOI:** 10.3390/s22072455

**Published:** 2022-03-23

**Authors:** Guo-Hong Chen, Jie Ni, Zhuo Chen, Hao Huang, Yun-Lei Sun, Wai Hung Ip, Kai Leung Yung

**Affiliations:** 1School of Information and Electrical Engineering, Zhejiang University City College, 51 Huzhou Street, Hangzhou 310015, China; chenguohong@zucc.edu.cn (G.-H.C.); nij@zucc.edu.cn (J.N.); chenz@zucc.edu.cn (Z.C.); 2Hubei Key Laboratory of Ferro- & Piezoelectric Materials and Devices, Faculty of Physics and Electronic Science, Hubei University, 368 Youyi Street, Wuhan 430062, China; 3Key Laboratory of Wireless Sensor Network & Communication, Shanghai Institute of Microsystem and Information Technology, Chinese Academy of Sciences, 865 Changning Road, Shanghai 200050, China; 4Department of Industrial and Systems Engineering, The Hong Kong Polytechnic University, Hong Kong, China; wh.ip@polyu.edu.hk (W.H.I.); kl.yung@polyu.edu.hk (K.L.Y.)

**Keywords:** pavement distress, feature combination, CNN

## Abstract

Aiming at the demand for rapid detection of highway pavement damage, many deep learning methods based on convolutional neural networks (CNNs) have been developed. However, CNN methods with raw image data require a high-performance hardware configuration and cost machine time. To reduce machine time and to apply the detection methods in common scenarios, the CNN structure with preprocessed image data needs to be simplified. In this work, a detection method based on a CNN and the combination of the grayscale and histogram of oriented gradients (HOG) features is proposed. First, the Gamma correction was employed to highlight the grayscale distribution of the damage area, which compresses the space of normal pavement. The preprocessed image was then divided into several unit cells, whose grayscale and HOG were calculated, respectively. The grayscale and HOG of each unit cell were combined to construct the grayscale-weighted HOG (GHOG) feature patterns. These feature patterns were input to the CNN with a specific structure and parameters. The trained indices suggested that the performance of the GHOG-based method was significantly improved, compared with the traditional HOG-based method. Furthermore, the GHOG-feature-based CNN technique exhibited flexibility and effectiveness under the same accuracy, in comparison to those deep learning techniques that directly deal with raw data. Since the grayscale has a definite physical meaning, the present detection method possesses a potential application for the further detection of damage details in the future.

## 1. Introduction

As highway infrastructure is developing rapidly currently, the pavement management standards generate the need for frequent inspection and maintenance. Distress detection and pavement diagnosis of highway infrastructure are key to guaranteeing their permanent availability. To improve the management of highways, it is efficient to carry out research and extend fast detection techniques [1,2].

Since image sensor technology has been developing in recent years, imaging speed has increased significantly, and imaging quality has also been improved. For instance, a global shutter camera can realize fast imaging, which has evident advantages in cost compared to ground penetrating radar or laser systems. This progress lays the foundation for road assessment and diagnosis based on machine vision. Therefore, image recognition technology for road damage can be achieved based on deep learning detection methods, instead of traditional image processing methods [3,4,5]. In detail, the threshold technique, contour detection, and frequency domain analysis, combined with specific geometric features, are employed to detect potholesand cracks. Koch and Brilakis proposed an automatic detection method for potholes in asphalt pavement, according to the shadow, shape, and surface texture [6]. Machine-learning-based methods first acquire the elementary features of potholes or cracks by a specific feature extraction method and train a classifier, such as support vector machine (SVM), k nearest neighbor, neural networks, and random forests, to recognize and classify images. Detection methods based on deep learning can extract features by automatic learning. Among them, the convolutional neural network (CNN) performs quite well in the machine vision field. Thus, several road damage detection methods based on CNNs have been proposed and demonstrated to have good effects [7,8,9], including VGG-16, ResNet, and DenseNet. Xiang et al. proposed a novel pavement crack detection method based on an end-to-end trainable deep convolution neural network [10]. The authors built the network using the encoder–decoder architecture and adopted a pyramid module to exploit global context information for the complex topology structure of cracks. However, training with high-resolution (HD) images, which contain much more information about the pavement, requires plenty of memory and computing power. So far, there are three main solutions as follows. The first one is to reduce the size of the input images, which may result in the loss of key detail features [11]. The second is to cut the HD images into several small pieces, which will then be used to train a specific CNN [12]. This solution, which requires much machine time, is inapplicable to those cases with high real-time requirements. The third one is to locate the target of the high-resolution images before training the neural networks [13]. Nevertheless, there are still some problems to be studied before a practical application. Compressing the raw data by these methods always leads to the loss of crucial details. On the other hand, CNNs could be optimized by considering a priori knowledge. The pavements with damage show geometrical features and a gray gradient characteristic. For instance, cracks always have an irregular, thin, and long shape, and the grayscale of the damage area is usually lower than the normal one. Under the limitation of the training set, it would not be easy to learn all these features. Combining the deep learning of CNNs with these features, it would be more effective and accurate for pavement damage detection. For example, Yousaf et al. extracted image features via the scale-invariant feature transform method and detected potholes in these images by the SVM classifier [14]. Qu et al. proposed a deeply supervised CNN with multiscale feature fusion, which introduced high-level features directly to the low-level features at different convolutional stages [15]. Li et al. propose an integrated system for the automatic extraction of pavement cracks based on progressive curvilinear structure filtering and optimized segmentation techniques [16]. Chen et al. proposed a second-order directional derivative to characterize the directional valley-like structure of cracks, in which the multi-scale Hessian structure was first proposed to analytically adapt to the direction and valley of cracking in the Gaussian scale space [17].

In this paper, we propose a highway pavement damage detection method based on a CNN and a combination of the grayscale and histograms of oriented gradients (HOGs), by considering various image processing techniques. Since road damage is always accompanied by local dents, we compressed the obtained HD images by removing these irrelevant pixels at the first stage, according to the grayscale change and edge shape. Then, we constructed the feature vector (feature pattern), based on the combined feature of the grayscale and histograms of oriented gradients, namely grayscale-weighted histograms of oriented gradients (GHOGs). The feature pattern, whose dimensions were dramatically reduced, contained the common features of the pavement damage. Finally, we input the feature pattern to build the CNN deep learning model for highway pavement damage detection.

## 2. Methods

### 2.1. Image Preprocessing

Before detecting the pavement damage, it is necessary to locate the target according to the common features, due to the noise signals caused by the device or the external environment during the data collection. As for highway pavement damage, there are cracks, potholes, and patches with various geometric shapes. Since highway pavement always shows a uniform color, the damage area is thus a set of pixels with relatively small grayscale values with respect to the background. The histograms of the pixels’ grayscale basically obey a Gaussian distribution, whose peak value corresponds to the overall status of the pavement [18],
(1)$$H\left(i\right)=a\exp\left(-{\left(i-u\right)}^{2}/\left(2\sigma^{2}\right)\right)+E\left(i\right),$$
where E(i) is the random noise. Therefore, the pavement images will first be dealt with by the Gamma correction, to raise the brightness level and to extend the dynamic range of the area with small grayscale values. Simultaneously, the grayscale range of the background, whose grayscale values are usually larger than those of the target area, is narrowed down by the Gamma correction,
(2)$$I\left(x,y\right)\to I{\left(x,y\right)}^{\gamma }.$$

Figure 1 shows the Gamma correction curves for different γ coefficients. As shown, the dashed line represents γ<1, when the dynamic range of the area with small grayscale values is extended and the contrast ratio is enhanced as well [19]. According to the histograms of the grayscale of the background pavement, a suitable γ is chosen to transfer the grayscale range of the damage area to similar ones as those of the background pavement, improving the detection effectiveness and efficiency. The γ value is determined by: (3)$$\gamma =\frac{\log(\left(I_{0}+0.5\right)/256)}{\log(\left(I_{p}+0.5\right)/256)},$$
where $I_{0}$ is the peak value before the transformation and $I_{p}$ is the one after the transformation. Figure 2 shows an image of highway pavement and its histograms of the grayscale before and after the Gamma correction.

### 2.2. Feature Combination of the Grayscale and HOG

#### 2.2.1. Calculation of the HOG

An image is first cut into $N_{c}\times N_{c}$ cells, and the HOG of each unit cell is then calculated. By using the Sobel operator, the vertical and horizontal gradients of the grayscale function can be obtained, which has filtering effects and speeds up the calculation. When the kernel size of the Sobel operator is 3, the oriented gradient can be obtained by: (4)$$G_{x}\left(x,y\right)=\left[\begin{array}{ccc} -1 & 0 & 1\\ -2 & 0 & 2\\ -1 & 0 & 1 \end{array}\right]*I\left(x,y\right)$$
and: (5)$$G_{y}\left(x,y\right)=\left[\begin{array}{ccc} -1 & -2 & -1\\ 0 & 0 & 0\\ 1 & 2 & 1 \end{array}\right]*I\left(x,y\right)$$
with the image function I(x,y). Thus, the amplitude and argument of the gradient are: (6)$$G\left(x,y\right)=\sqrt{G_{x}{\left(x,y\right)}^{2}+G_{y}{\left(x,y\right)}^{2}}$$
(7)$$\theta \left(x,y\right)=\tan^{-1}\left(\frac{G_{y}\left(x,y\right)}{G_{x}\left(x,y\right)}\right),$$
After the gradient of every cell is calculated, we then statistically analyzed the direction of the gradient based on the argument [20]. The range [0${}^{\circ }$, 180${}^{\circ }$] was divided equally into 9 directions, which serve as 9 basis vectors. Therefore, the gradient of each cell can be represented as a 9-dimensional vector or histogram $h_{cell}$. Figure 3 shows a typical histogram of oriented gradients for one cell. Four neighbor cells form a bigger unit, namely a block, whose histogram of the gradient direction is a series connection of the feature vectors of these four cell. Since the local amplitude of the gradient depends on the light illumination, and could vary in a wide range, it is reasonable to normalize the local gradient first.

#### 2.2.2. Grayscale-Weighted Histogram of Gradients

To improve the detection effect, it is efficient to combine different features of the specific object [21]. Compared with only extracting the HOG, the combination of the grayscale and HOG obviously contains more information. Therefore, we combined the grayscale and HOG of each cell and obtained the grayscale-weighted histogram of oriented gradients. In detail, we first calculated the average grayscale value of each unit cell $av_{cell}$. Then, the HOG of each unit cell was weighted by $av_{cell}$ to obtain the weighted feature vector [22]: (8)$$Fgg=h_{cell}\times \left(255-av_{cell}\right).$$

The Fgg of the four neighbor unit cells were connected to form a feature matrix, namely the GHOG image of a block. In this way, one can build a feature vector pattern of the whole image. If the raw image has N×N pixels, a unit cell has $N_{c}\times N_{c}$, a block has 4 unit cells, and the overlap between two neighboring blocks has 1 cell, there will be an $N_{g}\times N_{g}$ feature vector pattern with 36 channels, here: (9)$$N_{g}=\mathrm{mod}\left(\frac{N}{N_{c}}\right)-1.$$

Figure 4 shows a 160×160 image in the dataset and its corresponding Fgg feature vector pattern, with parameters $N_{c}=8$ and $N_{g}=19$.

### 2.3. Design of the CNN Classifier

Usually, the feature vectors of all the blocks in one image are connected to form a feature descriptor, which will then be detected and classified via SVM methods [14]. However, the two-dimensional structure of the blocks is not taken into account by these existing methods. Moreover, the feature vectors obtained by these method are too long to achieve ideal results [23]. The GHOG feature vector pattern keeps the two-dimensional structure of the blocks. Therefore, it might achieve the desired effect of detecting the GHOG pattern by the CNN method. In this paper, we input the GHOG image with dimensions (19, 19, 36) into the CNN as the input layer, and the basic frame of our CNN model is shown in Figure 5, which contains four convolutional layers with activation function ReLU, kernel size = (3, 3), and strides = (1, 1). Each convolutional layer is followed by one pooling layer with max-pooling sampling and strides = (1, 1) and one fully connected layer with the sigmoid and ReLU activation functions. The detailed information of the layers and parameters is displayed in Table 1.

### 2.4. Dataset and Performance Indices

The GAPs v2 dataset is the most extensive dataset in the pavement distress domain, which provides standardized, high-quality images [24]. The data of pavement damage, collected by a Pulnix TM2030 camera on an S.T.I.E.R. car, contains cracks, potholes, open joints, applied patches, and inlaid patches. GAPs v2 has a subdataset of image samples named 50 k, which includes a training set with 50,000 samples and three testing sets with 10,000 samples, named the validation set, validation–test set, and test set. These datasets offer a Python interface module for data downloading and training. Referring to the Chinese Standard Multifunctional High-speed Highway Condition Monitor (GB/T 26764-2011), Precision, recall, and F1 are introduced as the performance indices. Pavement damage detection is a binary classification, in which damaged pavement is a positive sample, while normal pavement is a negative one. Usually, TP represents the number of correct detections of pavement damage; TN is the number of correct detections of normal pavement; FP is the number of incorrect detections as damaged pavement; FN is the number of incorrect detections as normal pavement. The F-score, taking both precision and recall of the model into consideration, is the harmonic average of these two indices,
(10)$$F_{\beta }=\frac{(\beta^{2}+1)\times Precision\times Recall}{\beta^{2}\times Precision+Recall}.$$

When β>1, the recall has a bigger weight, when while β<1, the precision has a bigger weight. This formula gives the value of the F1 index if β=1.

## 3. Results and Discussion

Experiments were conducted on the subdataset 50 k of GAPs v2, 80% of which was used to train the model and the rest for testing. First, we studied the HOG feature pattern and GHOG feature pattern individually. Table 2 shows the performance indices for different feature patterns, where GHOG-1 represents the case in which the peak value became 0.6 after the Gamma correction and GHOG-2 represents that the peak value became 0.8. Compared with the HOG feature pattern, all the indices for the GHOG method were significantly enhanced. Moreover, GHOG-1 and GHOG-2 had no significant difference, indicating that the γ value had limited influence on the GHOG model.

Furthermore, we conducted an experiment on the training set of 50 k, to train our CNN model with a learning rate of 0.0001, 100 iterations, and a batch size of 25, to test the proposed model on the three test sets of 50 k. The obtained accuracy and loss are shown in Figure 6. As shown, for the first 20 training times, the accuracy and loss both changed rapidly, while after training 60 times, they gradually converged to 1.0 and 0.0, respectively.

Since the number of negative samples was much larger than that of the positive samples, it is reasonable to evaluate the credibility of our model for pavement detection by the precision–recall (PR) curve. For the application of pavement damage detection, images of pavement damage should be recognized as much as possible, which could be achieved by dropping the threshold value for positive samples. Nevertheless, images of normal pavement would be misdiagnosed as damaged pavement. Accordingly, the precision would tend to be smaller, while the recall would tend to be larger. In practice, by adjusting the threshold value of the positive samples, one can find an optimal solution from the PR curve, depending on the individual situations. Here, we show the PR curves of our trained model for the three testing subdatasets of GAPs v2 in Figure 7.

## 4. Conclusions

In summary, we proposed a highway pavement damage detection method based on a CNN and the combination of the grayscale and HOG. By image preprocessing of the Gamma correction and extracting the grayscale and HOG of the raw data, we obtained the grayscale-weighted HOG feature patterns first. These GHOG feature patterns were then input into the designed CNN classifier. The trained indices showed that the performance of the GHOG-based detection method was significantly improved, compared with the HOG-based one. On the other hand, the GHOG feature-based CNN technique exhibited flexibility and effectiveness under the same accuracy in the training and application, in comparison to those deep learning techniques that directly deal with raw data. Since the grayscale of pavement images have a definite physical meaning and are closely related to the damage types, it is expected that the GHOG-based method can be applied to further detection of damage details and to optimize the structure of CNNs in the future.

## Figures and Tables

**Figure 1 sensors-22-02455-f001:**
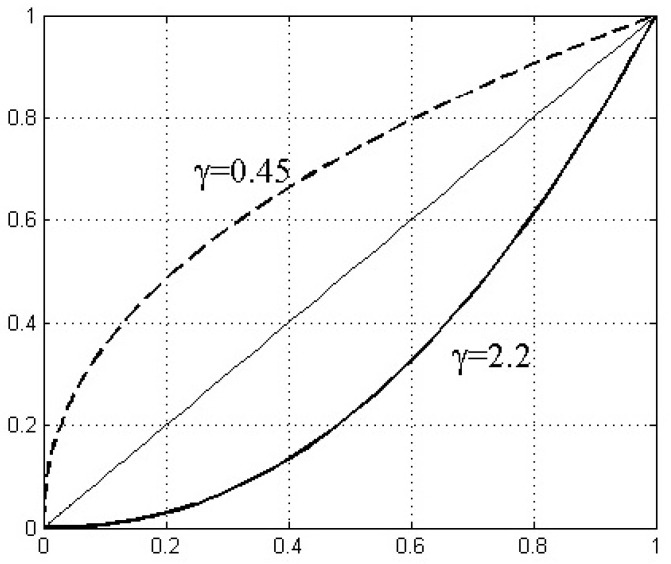
Gamma correction curves with various γ values.

**Figure 2 sensors-22-02455-f002:**
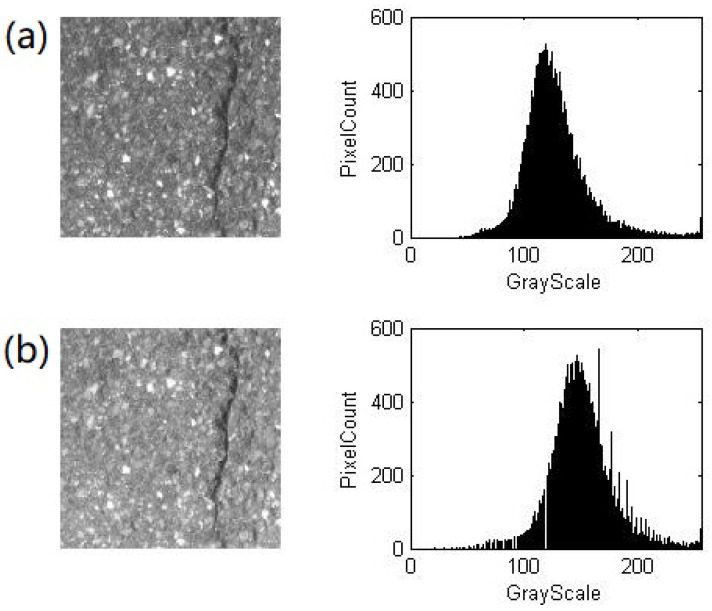
An image of highway pavement and its histograms of the grayscale (**a**) before and (**b**) after the Gamma correction.

**Figure 3 sensors-22-02455-f003:**
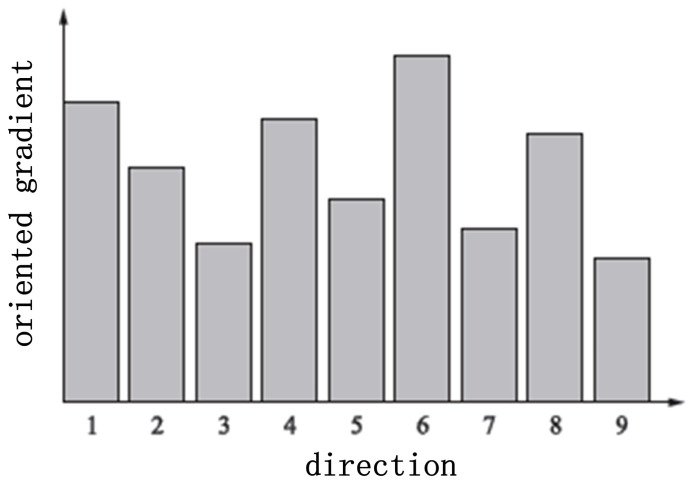
A typical histogram of oriented gradients for one cell.

**Figure 4 sensors-22-02455-f004:**
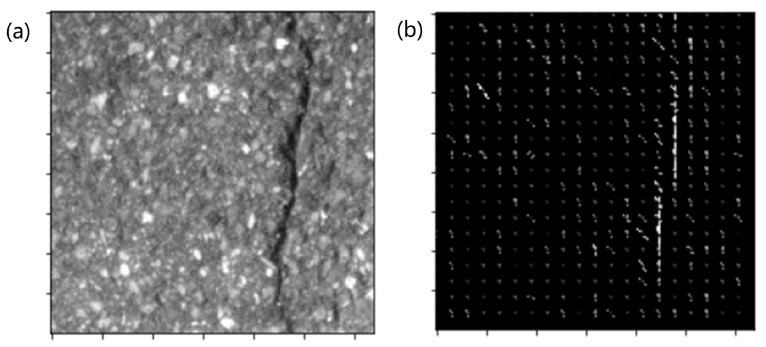
(**a**) Raw image and (**b**) its Fgg feature pattern.

**Figure 5 sensors-22-02455-f005:**
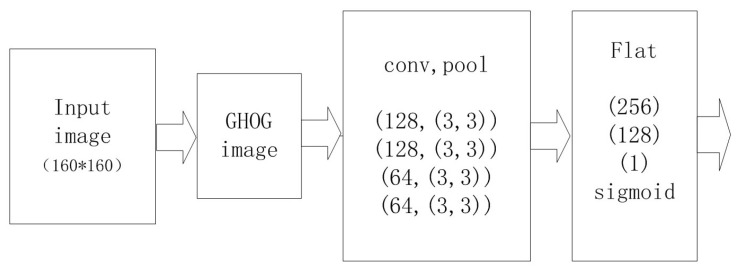
Structure of the CNN model in our research.

**Figure 6 sensors-22-02455-f006:**
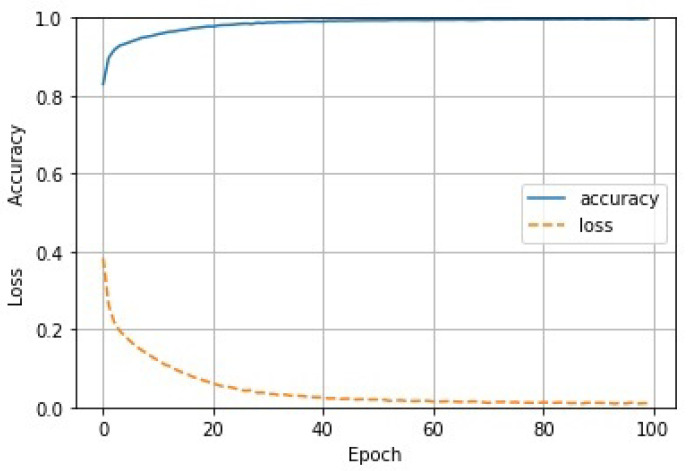
Accuracy and loss during the training of our model.

**Figure 7 sensors-22-02455-f007:**
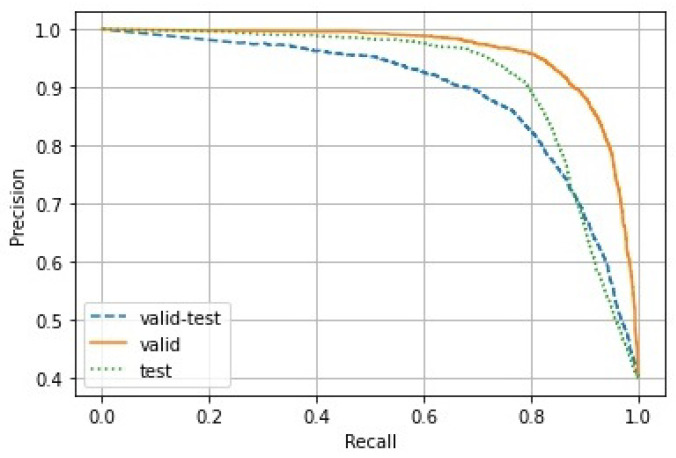
PR curves for the three subdatasets, validation–test, validation, and test.

**Table 1 sensors-22-02455-t001:** The detailed information of the layers and parameters in our research.

Layers	Output (Dimensions)	Parameters
conv2d_4(Conv2D)	(None, 17, 17, 128)	41,600
max_pooling2d_4(MaxPooling2D)	(None, 16, 16 128)	0
dropout_6(Dropout)	(None, 16, 16, 128)	0
conv2d_5(Conv2D)	(None, 14, 14, 128)	147,584
max_pooling2d_5(MaxPooling2D)	(None, 13, 13, 128)	0
dropout_7(Dropout)	(None, 13, 13, 128)	0
conv2d_6(Conv2D)	(None, 12, 12, 64)	32,832
max_pooling2d_6(MaxPooling2D)	(None, 11, 11, 64)	0
dropout_8(Dropout)	(None, 11, 11, 64)	0
conv2d_7(Conv2D)	(None, 10, 10, 64)	16,448
max_pooling2d_7(MaxPooling2D)	(None, 9, 9, 64)	0
dropout_9(Dropout)	(None, 9, 9, 64)	0
flatten_1(Flatten)	(None, 5184)	0
dense_3(Dense)	(None, 256)	1,327,360
activation_3(Activation)	(None, 256)	0
dropout_10(Dropout)	(None, 256)	0
dense_4(Dense)	(None, 128)	32,896
activation_4(Activation)	(None, 128)	0
dropout_11(Dropout)	(None, 128)	0
dense_5(Dense)	(None, 1)	129
activation_5(Activation)	(None, 1)	0
Total parameters: 1,598,849;		
Training parameters: 1,598,849;		
Non-trainable parameters: 0.		

**Table 2 sensors-22-02455-t002:** The performance indices for the HOG and GHOG feature patterns. GHOG-1 and GHOG-2 correspond to two different Gamma corrections.

Feature Pattern	Accuracy	Precision	Recall	F1
HOG	0.8395	0.8466	0.7400	0.7981
GHOG-1	0.9500	0.9476	0.9286	0.9380
GHOG-2	0.9463	0.9547	0.9110	0.9385

## Data Availability

The data presented in this study are available in the article.

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
