# Peer review of "Detection of Highway Pavement Damage Based on a CNN Using Grayscale and HOG Features"

_sensors, 2022, doi:10.3390/s22072455_

Round 1

Reviewer 1 Report

In the paper the authors presented a highway pavement damage detection method based on convolutional neural network and combination of grayscale and histogram of oriented gradient. After Gamma correction of the image and extracting grayscale and HOG of raw data the grayscale-weighted HOG feature patterns are obtained which, as the next step, are an input into the designed CNN classifier.

The paper is interesting, however, I suggest to take into account the following remarks:

  1. an extensive proofreading is necessary (misspelings, e.g. line 16, grammar mistakes, e.g. line 69)
  2. the introduction and bibliography should be more extensive
  3. in some figures (e.g. 2,4) the labeling of the axes is missing
  4. for clearer presentation of the results in some figures (e.g. 2,4) introduction of (a), (b) would be beneficial
  5. table 2 should not be at the beginning of the section 3
  6. conclusions are more or less the same as abstract. Please, emphasise here the novelty and advantages of the conducted study.

Reviewer 2 Report

The author presented CNN based method to detect highway pavement damages. Overall paper is interesting, but I have the following concerns:

1) Abstract is directly discussing the CNN and HOG, The authors should change the abstract entirely based on  " Problem definition, lacking existing approaches, proposed approach, outcomes of research"

2) The introduction is insufficient, in the overall introduction, I have found just one latest paper, authors should discuss the latest approaches like from:

"A critical review and comparative study on image segmentation-based techniques for pavement crack detection"

"Use of Parallel ResNet for High-Performance Pavement Crack Detection and Measurement"

3) It is very strange to conclude the deep learning-based paper in just 18 references overall. 

4) Table 1 is wrong or not clearly presented, please clarify  (None, 17, 17, 128) what is none? what is 17? what is 128? 

5) If this 17,17, 128 is 17×17× 128 then it is very strange that when the input image of 160×160 is provided to a convolutional layer how can it output 17×17× 128 which is so small. 

6) I am confused why this CNN architecture is picked? Did you experiment with the same data with VGG-16?, ResNet? DenseNet? what are the results?

7) I think the famous networks like VGG/ResNet/DenseNet will provide direct good results without HOG features, I think Authors should experiment with the enhanced images (shown in upper left of  Figure 2) without HOG, and present the results in comparison with HOG, CHOG-1, and CHOG-2 in Table 2

Note: It does not matter if VGG/ResNet/DenseNet provide better results, still the paper will be in good form if these things are presented in a good way 

Round 2

Reviewer 2 Report

The authors responded to most of the concerns. I recommend acceptance of the paper in its current form.